# OpenReview forum: "CoS: Enhancing Personalization and Mitigating Bias with Context Steering"
_NeurIPS.cc/2024/Workshop/SafeGenAi — SafeGenAi Poster_

### Official Review · Reviewer_wmVV · 2024-10-09
**Peer Review for CoS: Enhancing Personalization and Mitigating Bias with Context Steering**

**Rating:** 6
**Confidence:** 3

**Review:**

## Strengths
1. **Originality**: the paper introduces Context Steering (CoS) a simple training-free method that can be applied to LLMs easily to mitigate potential bias
2. **Societal implications**: The paper tackles a significant issue by proposing a solution to reduce bias in LLMs. This is crucial given the growing concern over fairness and ethics in AI, which is a timely contribution

## Areas for improvement
1. **Discussion on limitations and future research directions**: What are the potential limitations of CoS? For example, would CoS impact LLM's ability to infer? How should future studies address the tradeoff between bias mitigation and inference ability?
2. **Structuring**: the tables and visualizations could be placed closer to the relevant texts. For example, Table 1 results were discussed at the bottom of page 4 while the table itself is on top of page 3, which affects readability

---

### Official Review · Reviewer_HiSg · 2024-10-09
**Novel use of context for personalization and Bayesian inference to recover context**

**Rating:** 8
**Confidence:** 3

**Review:**

This paper introduces a new way to utilize context, allowing for a continuous scale of context that can increase or decrease its weight to tailor LLM responses. The authors show that this can be inverted, using Bayesian inference to recover the likelihood of specific contexts being used conditional on the response.

Pros:
- The authors demonstrate how their approach can be used for personalisation of LLM models
- The authors demonstrate how their approach can be used to detect bias at rates similar to humans

Cons:
- It's not clear how important this ability to tune the strength of personalisation is. The authors show that the LLM can generate more personalized recommendations. It would be helpful for authors to also demonstrate that people with different preferences (ex/ those that enjoy one genre of movie over the other) actually prefer the more personalized recommendations (potentially up to some $\lambda$ beyond which it is too much)
- It would be nice to conduct some statistical inference when comparing benchmarks to understand whether we can actually distinguish the improvements of the model versus other appproaches

More broadly speaking, there are lots of situations where determining context is useful. Most classification tasks could be thought of in this way. A more systematic exploration of when direct prompting vs. the CoS-Bayesian inference strategy leads to better accuracy would be interesting. This aspect of the paper seems like the largest contribution to me: an alternative way to ask LLMs for answers.